# ScaleUp: middleware for intelligent environments

Daniyal Alghazzawi, Ghadah Aldabbagh and Abdullah Saad AL-Malaise AL-Ghamdi

Faculty of Computing and Information Technology, King Abdulaziz University, Jeddah, Saudi Arabia

## ABSTRACT

The development of the Internet of Things (IoT) expands to an ultra-large-scale, which provides numerous services across different domains and environments. The use of middleware eases application development by providing the necessary functional capability. This paper presents a new form of middleware for controlling smart devices installed in an intelligent environment. This new form of middleware functioned seamlessly with any manufacturer API or bespoke controller program. It acts as an all-encompassing top layer of middleware in an intelligent environment control system capable of handling numerous different types of devices simultaneously. This protected de-synchronization of data stored in clone devices. It showed that in this middleware, the clone devices were regularly synchronized with their original master such as locally stored representations were continuously updated with the known true state values.

## INTRODUCTION

Architectures for intelligent environments typically require some form of necessary middleware layer, enabling installed hardware (smart devices) and software, (agents or controller programs), to link and operate together. Various types of middleware are created to perform this crucial role, such as Universal Plug and Play (UPnP), Web Services, and bespoke mechanisms written by device manufacturers (*Tan et al., 2015*).

Middleware may vary in terms of functionality, with different strengths and weaknesses. These strengths and weaknesses largely depend on their deployment conditions. This is challenging for the programmers that adopt unenviable measures when designing control systems, particularly for complex smart devices or intelligent environments. Selecting a particular mechanism requires trade-off of resources or functionality to a certain extent (*Rafique et al., 2015*). Some bespoke control system middleware designs help recover functionality loss by reproducing components that help to extend the chosen mechanism. Alternatively, programmers could implement an architecture where multiple independent middleware layers run simultaneously within a single control system. Both of these solutions naturally require a greater coding effort to implement the system, which itself is a lengthy program processing than the default implementation.



Corresponding author
Daniyal Alghazzawi,
dghazzawi@kau.edu.sa

A potential risk of data synchronization also prevails within the system (*Bruneo, Puliafito & Scarpa, 2006*). For instance, the device state variability; updated using one middleware system, cannot be reflected by all the alternative mechanisms used. If their state data became desynchronized, smart devices controlled by agents may exhibit erroneous or undesirable behaviours, which could prove costly and dangerous. These issues also emerge due to hardware action request response times. If the new state data needs to be shared amongst several components within the middleware (e.g., to synchronise the system), this additional processing could be slower to changing events within the environment. This could frustrate users when updating their device and makes operating agents more apparent within an intelligent environment; for example, walking into a dark room and needing to wait for the lights to switch on.

This paper highlights work performed by an ongoing international research project (ScaleUp project) investigating methods for scaling up intelligent environments. As a consequence of this research, a new middleware was developed, specifically designed to address the trade-off issue, when deciding between functionality provided by UPnP and Web Services implementations. The aim was to amalgamate distinctive beneficial features offered by the UPnP and Web Services approaches into a new middleware design. The new mechanism was also structured to protect intelligent environment control systems against data de-synchronization, and processing action requests in times at least on par with other existing middleware examples. This work was based on ongoing international collaboration between the University of Essex and King Abdulaziz University.

The core contribution of this study is an evaluation of both the state-of-the-art and state-of-the-practice in the middleware research offered by existing middleware products. Software engineers progressively utilize middleware for building distributed systems. Any research into ScaleUp project that avoids this trend will merely have confined effect. Therefore, this paper has analysed the effects of middleware on the software engineering research agenda. It is argued that requirements engineering techniques are required for focusing on non-functional requirements, as these impact the selection and use of middleware. Software architecture research can develop methods guiding students towards the selection of right middleware and integrating it so that it fulfils a series of non-functional requirements.

## LITERATURE REVIEW

### Related work

Currently, many off-the-shelf commercial devices possess inbuilt intelligent functionality (i.e., either contain or can be accessed in some way via a computer system) (*Yuan, Xiaolei & Yitao, 2019*). To facilitate usability of their products, hardware manufacturers include pre-installed middleware designed for computer control. Previously, control systems were device-centric; for example, the control system for washing machine cycle programming. However, with the widespread use of networking platforms such as Ethernet, Wi-Fi, and Bluetooth, the interconnectivity between different devices began to emerge. Being proponents of Pervasive Computer Science concepts, such as the

Internet-of-Things (*Gubbi et al., 2013*), several hardware manufacturers chose to implement middleware for their products around established common frameworks, such as UPnP.

This was due to a belief that by adopting common middleware architecture, customers would use smart devices created by rival companies, within their proprietary personal home networks. Though, this concept is yet to be turned into a mainstream commercial reality. Various larger multinational hardware manufacturers, such as Apple or Sony are adopting more protectionist marketing strategies, creating interconnectivity between their products and services. For example, Apple iWatch smart device release will only work when paired with an Apple iPhone 5 or newer, though it is not compatible with any other older iPhone models or any rival smart-phone handset (*Almusaylim & Noor, 2019*). Such smart devices operate using completely bespoke proprietary middleware systems, designed solely by a particular company.

The current literature shows that the use of middleware in IoT is limited (*Farahzadi et al., 2018*). *Ngu et al. (2016)* have designed an IoT application for real-time prediction of blood alcohol content based on smartwatch sensor data. The study has conducted a survey on the competencies of the current IoT middleware. In addition, the challenges and the enablers associated to the IoT middleware were presented in order to embrace the heterogeneity of IoT devices. *Fremantle & Scott (2017)* have utilized a structured search approach for identifying 54 particular IoT middleware frameworks and examined the security frameworks associated to each middleware. A total of 12 requirements (integrity and confidentiality, access control, consent, policy-based security, authentication, federated identity, and device identity) were used from the first stage for validating the competencies of each system.

*Elkhodr, Shahrestani & Cheung (2016)* have accounted middleware for the emerging attributes such as seamless communications, lightweight aspects, and mobile across different heterogeneous networks and domains. It involves a context-adaptive technique, which allows the user for managing the location information shared by things on the basis of policy enforcement and context-aware mechanism. This mechanism accounted both the preferences and informed consent of a user. *Jyothi (2016)* has managed data volumes and supported semantic modeling in the open issues, specifically managing the crowd sourcing of different domains. There is a scope for research work in order to make a generic IoT-middleware system, which is relevant across all regions by making all the functional aspects reusable and can be included as enabler to the middleware system.

*Razzaque et al. (2015)* survey showed that the use of middleware assists in the development through its integration of heterogeneous communication and computing devices. It also states that middleware provides interoperability support for application across different devices and services. *Jeon & Jung (2017)* have revealed that the average request rate elevated by 25 percent compared to Californium, which is a middleware for effective association in IoT environments with vigorous performance, a power consumption reduced by up to 68% and an average response time reduced by 90% when resource management was utilized. Lastly, the latency and power consumption of IoT devices can be reduced by the proposed platform. According to *Cruz et al. (2018)*, an

important role is played by middleware as it is accountable for covering the intelligence part in IoT, which make decisions, allow them to communicate, and integrate data from devices on the basis of data collected. Afterward, a reference architecture model was investigated for IoT middleware based on IoT platform requirements, which detail the effective operational approaches of each proposed module, and proposed fundamental security features for this software type. *Zhao et al. (2018)* have proposed a stack of support-communication-computing for integrating effective open-source projects in order to devise techniques for allowing sufficient uniform human-thing associations, and developing implementation foundations for cutting-edge technologies including semantic reasoning and fog computing.

Due to the lack of common frameworks adoption for building complex intelligent environments or devices, the addition of a bespoke middleware layer is critical. This layer is designed to collectively handle the respective control systems of each installed smart device type. This extra layer of middleware wraps the individual bespoke device controller programs into a single API, representing the entire intelligent environment. This is also created by different Computer Science projects, which create middleware for intelligent environments, linking their creations into bespoke device controllers. Although all might concentrate on the same area of the control system architecture, the actual functionality can have a broad design. For example, *Rivera & van der Meulen (2013)* created their bespoke Gaia infrastructure to allow distributed collections of smart objects and environments, represented and accessed via an interface. Later, *Perera et al. (2014)* chose to integrate a control system for a robot into their intelligent environment middleware, allowing it to seamlessly access sensors and actuators within the smart space. In their OpenCOPI middleware, *Bazzani et al. (2012)* utilized a web ontology language based around semantic web services to create a mediator regulating access between ubiquitous applications and available service providers. *Echelon (2015)* also used web Services as the base for their aWESoME infrastructure, which in addition to being an intelligent environment controller, focused on promoting "energy savings" by consuming low power in its operations. Finally, *Phidgets (2015)* used a context-aware multi-agent system as their middleware base for controlling ambient intelligence exhibited within an environment.

Different middleware platforms were discussed on these criteria for networked robotic systems (*Mohamed, Al-Jaroodi & Jawhar, 2009*). Majority of the middleware platforms have varied objectives including reusability, development process, self-discovery, self-configuration, supporting QoS, flexibility, and integration. In addition, several middleware platforms were discussed in the study for networked robotic systems such as self-adaptation, discovery and higher-level abstractions, collaborations support, and other advanced characteristics for integration.

*Rodriguez-Molina & Kammen (2018)* have apparently demonstrated the collection of services demonstrated by community of researchers, developers, and scientists. Furthermore, middleware solution utilizes an API that explains how services were accessed from both the applications and the hardware that has been embedded for a Smart Grid-like deployment. Moreover, the authors have indicated that boundaries exist between the hardware located and the network, which are compliant with a standard demonstrating the

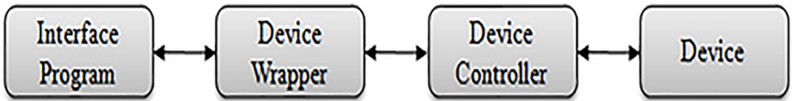

**Figure 1 An example of a two-tiered middleware architecture for controlling a smart device.**

sub-systems of the software. *Vikash, Mishra & Varma (2020)* have conducted a survey on middle-wares of WSNs towards IoT in offering a comparative view of different middle-wares and how this middleware technology can be utilized for implementing several issues emerging in the development of IoT applications. *Rodríguez-Molina et al. (2017)* have utilized different aspects with several functionalities preferred as essential for a semantic middleware architecture conjoined with maritime operations including access to the application layer, context awareness, and device and service registration. On the contrary, *Rodríguez-Molina et al. (2017)* have interweaved other technologies with middleware such as acoustic networks and wireless communications. Under such circumstances, *Rodríguez-Molina et al. (2017)* have established an approach for interchanging information at the data level among independent maritime vehicles, which is of significant importance as the required information will have to be defined, along with the size of transferred data. *Rodríguez-Molina et al. (2017)* have forwarded the Maritime Data Transfer Protocol for interchanging standardized aspects of information at the data level for maritime independent maritime vehicles, and the procedures that are needed for information interchange.

## Techniques/Protocols/Paradigms used in the development of proposed work

Figure 1 highlights the integration of the additional middleware layer within the control system architecture using the 'Device Wrapper' node. Without this layer of middleware, it would be necessary first to learn the use of interface and to control each smart device individually. This requires the use of multiple different programming languages, such as Java, Python, C, C++ or C#, and possibly OS-dependent software packages. For example, the University of Essex currently has two full-scale intelligent environment test-beds, each using a different style of additional middleware layer to amalgamate devices from various manufacturers. These include iClassroom and iSpace.

The iClassroom (*Román et al., 2002*) is an intelligent environment customised to resemble a university or school teaching room (Fig. 2). Most smart technologies are used directly to augment presentations or other teaching strategies to enhance student learning experiences. The middleware in the environment uses Web Services to wrap diverse collection of devices into a single common interfacing mechanism. The Web Services system uses Eclipse's Jetty Web Server (*Roalter, Kranz & Möller, 2010*) for its operations.

The iClassroom uses a centralized configuration, with all the middleware running from a single server. In terms of functionality, the main user interface is in bespoke website form hosted on the environment server. This server contains numerous hyperlinks to the Web Services used to control each device. The user interface can be loaded onto any standard

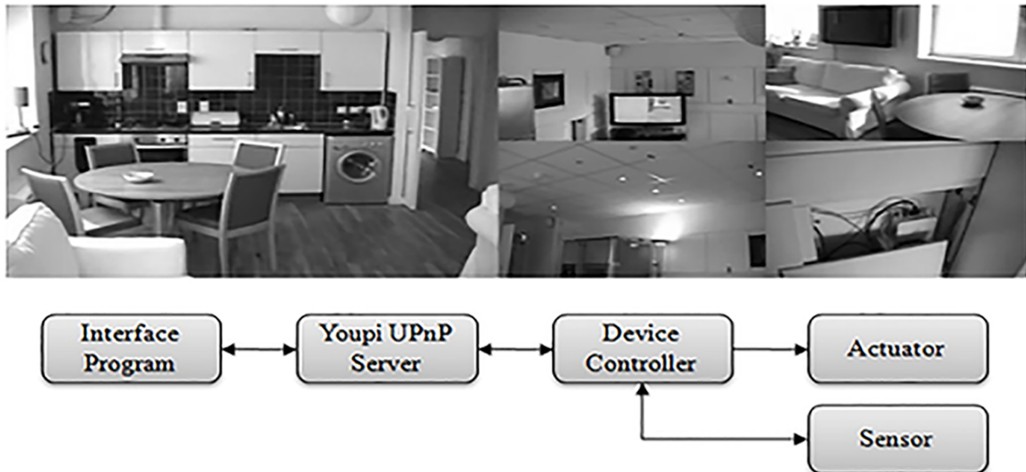

**Figure 2 The iClassroom intelligent environment and its middleware architecture.**

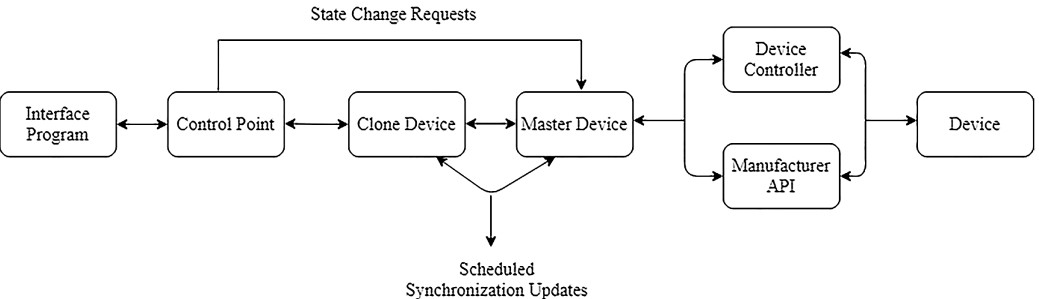

**Figure 3 The iSpace intelligent environment and its middleware architecture.**

browser application such as smart phones and tablets. Through the user interface, smart device in the environment can be connected, controlled, and monitored to the Web Services system from any remote location capable of accessing the iClassroom network.

The iSpace (*Lopes et al., 2014*; *Stavropoulos et al., 2013*) is an intelligent environment test-bed customized to resemble a typical household environment. The multi-roomed space includes a full-sized lounge/kitchen, study, bathroom, and bedroom, each connected by a central hallway. Unlike the iClassroom, most of the smart technologies in the iSpace are deliberately concealed within hollow walls and ceilings (Fig. 3). This provides unknowing visitors to space an initial impression that they are in a normal (i.e., non-augmented) environment. The middleware in this intelligent environment is based upon UPnP, with controllers for over sixty smart devices linked together into a single API. More specifically, Youpi UPnP stack (*Olaru, Florea & Seghrouchni, 2013*) was used to implement wrappers for each smart device.

The iSpace comprises a distributed configuration, with the middleware and control code for devices which split across several different computers, each connected via a common network. The Youpi UPnP wrappers allow each smart device to broadcast its existence

on the same network (*Dooley et al., 2011*). iSpace allows programmers to create a bespoke controller code to integrate the smart devices into their projects better. To discover them, programmers must create a Youpi UPnP control point within their code and perform a search. This can be either for a specific device or a general search that returns a set of every smart device discovered on the network. Using a returned smart device instance, a control program can then isolate specific state variables, actions and arguments to either monitor or modify the associated hardware. Using UPnP, user control programs can 'subscribe' to individual state variables within a smart device instance. The state of a subscribed variable changes can automatically flag an associated listener within the user's interface program.

For a large environment like the iSpace, implementation and maintenance of UPnP middleware require significant programming effort. To operate a wrapper for each smart device instance, a significant amount of information concerning hardware and its uses are required. Initially, each device type needed every state variable, action, and argument that can be individually declared. These were associated with dedicated UPnP state listeners to allow the correct functioning of the subscription system. Finally, it was necessary to individually initialize and start each instance of a smart device in the iSpace, which included assigning unique attributes such as names and UUIDs.

## THE ENVIRONMENT DEVELOPMENT KIT (EDK)

The new system is created from scratch and is not an extension or reconfiguration of a pre-existing architecture. Thereby, steps should be taken to address the synchronisation and resource allocation issues. This led to the creation of the Environment Development Kit (EDK), which was written using only standard Java SDK, with no extensions or third-party APIs. The EDK architecture design declared smart device present within an Intelligent Environment to be;

a) Discoverable on a network.
b) Subscribed to using listeners, which monitored for state change events.
c) Accessible and/or controllable via a common API.
d) Accessible and/or controllable via a set of auto-generated Web Services.
e) Assigned with a unique controller program instance or one collectively shared by a group.

The system operates in both a centralized or distributed context and is not OS dependent. Controllers assigned to individual or groups of devices could be updated using a 'hot-swap', without needing to restart any part of the system, (as long as the device is not being accessed at the time of modification). Additionally, the inventory of devices is not to be declared before the system could run, as new devices could be discovered and handled by the system at any time. The case is similar to various middleware previously designed for intelligent environments, such as UPnP.

A multicast communications system allows devices to be linked to agents and other user programs. Multicast is less dependable than alternative communication methods such

```
private final int COMMUNICATIONS_PORT = 4446;
private final int WEBSERVICES_PORT = 8000;
private final String CONTROLLER_CLASS = "Controller";
private final String CONTROLLER_PACKAGE = "devicecontroller";
private final String CONTROLLER_DIRECTORY = "controllers";
private final String NETWORK_ADDRESS = "230.0.0.1";
```

**Figure 4** The EDK middleware architecture.    

as TCP. It is because there is no guarantee that intended targets would receive the message broadcasted to a network. However, it is significantly much faster than TCP and can share message across multiple targets in a subscribed group simultaneously, which determines the main reasons behind its usage in this middleware system.

Similar multicast systems are in use in computer game virtual worlds, where the environment is divided into regions, each with their assigned group containing all the players, objects and other updatable information present in that area (*Simó-Ten et al., 2017*; *J. Eclipse, 2015*). When an entity moves from one region to another, it should also change its subscription to the corresponding multicast group while simultaneously leaving the previous one.

Users were able to access the EDK network, search for represented smart devices and send action requests to specific discovered instances via an inbuilt control point. Devices discovered by each control point instance were automatically 'cloned' and locally stored. Clone devices were exact copies of their originals, with the exception that they possessed no controller, so they could not directly interface with real hardware in the environment. Cloned devices received state updates from the original 'master' device. These synchronisation updates occur regularly. Typically, this requires an update several times a minute, although the interval period between messages could be increased or decreased according based on its importance for clones to return the most up-to-date state information possible.

For actuators, EDK control points bypass the clone representation and transmit a state change action request directly to the master device. Multicast communication used received response from the device indicating concerning the change of request wither its success or failure. However, if the new state was applied to the master device, then alteration of the local clone would occur during the next synchronisation update. To speed this process, the EDK was designed where changes are automatically and immediately prompted to a synchronisation update, temporarily overriding the scheduled system, (which had its internal timer reset). Figure 4 shows complete EDK system architecture for controlling a single, smart device via a third-party interface program.

Allowing users to create control points, the API provides a series of expandable classes and methods that are used with a broad range of bespoke sophistication for devices or environments. The EDK came packaged with classes for representing several basic actuator and sensor types, used by master device representations, and subsequently by control point instances when creating their clones. Furthermore, the system was equally capable of

```
BooleanLight light1 =
        new BooleanLight("Light1", "Power", "Light Mk 1");
light1.addStateChangeListener(sl);
```

**Figure 5** An example of communication and controller constants.

handling encountering unknown device types in an ad-hoc manner. This was achieved by creating a cloned instance of the discovered device using a generic class appropriate for representing its declared features and variables. Several different generic classes existed depending on the device, whether the device was an actuator or a sensor and the number variables it possessed. Because of this feature, control points did not require setup or inventory of an intelligent environment before being connected. Additionally, it was possible to use the EDK with intelligent environments based around both centralized and distributed architectures, as there was no requirement to host master devices and clones on the same computer to communicate.

## Implementing master devices

Three key components are required to exist on an EDK network;

1. A representation identifies the attributes of the device and declaring whether it is an actuator or sensor.
2. A gateway to the target EDK network from where the device will be accessible.
3. A controller connecting the virtual EDK representation with some counterpart hardware that exists in the real-world.

Together these components allow a "master" device to be created using the EDK API. Once deployed on the network, the virtual device representation would monitor or change the state of the real hardware-based upon received user instructions. It would also periodically broadcast details of itself and the current state of the hardware to the EDK network, which is received by any active control points and used to create or synchronize associated "clone" devices.

Master devices can be implemented and deployed remotely on one or several networked computers. Depending upon personal preference, a master device application could also be used to generate one or several different actuators and sensors, with no requirement of device type. Multiple different control programs can be imported into the same implementation, and if desired, a group of devices can either share or each is allocated their instance of the same controller.

### Step one: declaring constants

Firstly, several constants need to be declared to allow communication and different control systems to operate correctly. Figure 5 provide examples of these, where each attribute is explained further in the implementation process.

```
DimmableLight dimmer1 = new DimmableLight("Dimmer1",
        new String[] { "Power", "Brightness" }, "Dimmer Light Mk 1");
dimmer1.addStateChangeListener(sl);
```

**Figure 6  Creating a new single variable smart device.**

```
DeviceHub lighthub = new DeviceHub (communications);
lightHub.addDevice (light1, lightController);
lightHub.addDevice (dimmer1, dimmerController);
```

**Figure 7  Creating a new multi-variable smart device.**

### Step two: creating device instances

The EDK API contains several different classes that can be used to create new representations of individual master devices. The selection of the class for a given situation depends upon several factors, specifically, the number of state variables supported by the device along with its identification as an actuator or a sensor. For common device types, the EDK API contains several dedicated classes, many of which contain bespoke convenience methods that provide better control of specific state variables. For example, Figs. 6 and 7 show the creation of an instance of two different types of light emitting devices. Both types have a state variable "Power" declared, which controls whether the device is on or off. To support these actions, both the "BooleanLight" and "DimmableLight" classes contain two convenience methods "turnOn" and "turnoff" which allow the "Power" state variable to be set without the need to declare or process any new values directly. The "DimmableLight" class used in Fig. 7 additionally contains "getBrightness" and "setBrightness" methods, as compared to the declared "Brightness" state variable, used to control the emitted light level.

Without these convenience methods, the state variable name would need to be declared along with any new state value (if applicable), in a formatted String in order to perform the same function. If the device being used is uncommon or unknown for some reason, the EDK API generic series classes can be created. This creates representations based purely on the number of declared state variables. For actuators, these would be "SingleVariableActuator" and "MultiVariableActuator", whereas for sensors the appropriate generic classes would be "SingleVariableSensor" and "MultiVariableSensor". Each class contains several different constructors based on the available information of the created device. For instance, in Figs. 6 and 7, the constructor is provided with a name for the device instance, the supported state variables (provided in a string array for multi-variable devices, as seen in Fig. 7), and a description of the device itself. In this instance, all other required variables, such as a unique UUID, are allocated to the new device by the EDK API. Figures 6 and 7 also show how a "StateChangeListener" can be attached to individual master device instances, which flag whenever the value of any

```
InetAddress environmentAddress =
        InetAddress. getByName (NETWORK_ADDRESS);
Communications communications =
new Communications (environmentAddress, COMMUNICATIONS_PORT);
```

**Figure 8   Creating the communications system.**         

```
File controllerJARFile = new File (CONTROLLER_DIRECTORY +
        File.separator + "DeviceController.jar");
Controller lightController = new Controller (
     controllerJARFile, CONTROLLER_PACKAGE, CONTROLLER_CLASS);
```

**Figure 9   Loading device control systems.**         

supported state variable is changed either by some attached hardware or due to a user request.[1]

### Step three: adding communications

An instance of the "Communications" class must be created in order for master devices to be able to connect to an EDK network. Figure 8 shows how to do this using two of the variables from Fig. 5 mentioned earlier to supply values for the network address and communications port variables. The EDK uses multicast communication to allow master devices to send updates to any running control points that have joined the same group. As a consequence, it is important to ensure that the value used for the "NETWORK_ADDRESS" variable is a valid multicast address. It may also be necessary to open the value used for "COMMUNICATIONS_PORT" on firewalls, which may be blocking the sending or receiving of multicast communications packets on the network.

### Step four: loading device controllers

Control system programs for individual smart devices are created independently of the EDK. Once implemented, a device control system can be uploaded into a master device via the EDK API. Figure 9 shows how to do this. "DeviceController.jar" is the filename of the controller being uploaded, from the designated "CONTROLLER_DIRECTORY". To integrate the controller program with the master device, the EDK needs to know the package ("CONTROLLER_PACKAGE") and the name of the main class ("CONTROLLER_CLASS").

As before, examples of these values are provided in Fig. 5.

### Step five: creating a device processing hub

The next step is to create a hub to process devices. The "communications" variable of the "DeviceHub" constructor should be the same instance of the "Communications" class created back in Step Three. Once the hub is initialised, each of the smart device instances created in Step Two needs to be individually added. As these are master devices, they also

[1] Note: When implementing master devices, it is highly recommended that each instance is given unique name value. It should always be ensured that device instances always have different uuid values, which can be generated randomly using the Java SDK UUID class. Note: All supported state variable names must be declared when creating a new device instance. Several pre-formed device types are included within the EDK API, which often contain several convenience methods for performing device-specific actions. To use these methods, state variables must be declared using the specific names specified by the Javadoc information for the relevant class. Note: If an application is intended to deploy multiple master device instances, it is recommended to create each instance at this stage before continuing. Note: To prevent processing errors, individual state variables should never include the colon or tilde characters (i.e.: or ~) in their names or possible returnable values.

```
DeviceHub lightHub = new DeviceHub (communications);
lightHub.addDevice (light1, lightController);
lightHub.addDevice (dimmer1, dimmerController);
```

**Figure 10  Creating a hub and adding master devices.**

```
WebServer webServer = new Webserver (WEBSERVICES_PORT);
lightHub.addWebServices (webSERVER);
```

**Figure 11  Enabling the EDK web services.**

need to be associated with their respective controller programs, loaded during Step Four. Figure 10 provides an example of how to implement this for light devices.

The purpose of the "DeviceHub" class differs slightly depending upon whether it is used within an implementation for master devices or an EDK control point. For master devices, the hub liaises with the communication system created in Step Three, to have each of its stored devices access their associated controllers and perform a live update of their recorded state variables. Typically, this would involve each device accessing the real-world hardware to which its control system connects it. Once acquired, the up-to-date state variable information is then passed back to the communications system, where it is transmitted to the EDK network and used synchronizing any listening clone device representations.

### Step six: adding web services (optional)

Using the Web Services interface method with an EDK middleware implementation is optional, so this step can be ignored if desired. However, enabling the Web Services system only requires the code shown in Fig. 11 to be added after the creation of the device hub. The variable "WEBSERVICES_PORT" is the last of the declared constants create back in Step One, which in the case of this example (Fig. 5).

Once added to an instance of the "DeviceHub" class, the EDK mechanism will auto-generate a Web Services control interface for each of the declared smart devices and add them to the internal HTTP Server. Currently, two different acceptable commands are implemented for sensors (i.e., about and get), while three for actuators (i.e., about, get and set). The Web Services interface can be loaded using any standard Web Browser, including on most mobile devices, such as smartphones and tablet computers. The syntax for an EDK Web Service is naturally bespoke to each situation where it is used, but the basic URL structure is as follows:

**http://<ComputerIPAddress>:<WebServerPort>/<DeviceName>/<Command>**

So, based upon the "BooleanLight" and "DimmableLight" smart device examples used throughout this tutorial, some acceptable URLs would be:

http://127.0.0.1:8000/Light1/about

http://127.0.0.1:8000/Light1/get

http://127.0.0.1:8000/Light1/set?Power:1

http://127.0.0.1:8000/Dimmer1/get?Brightness

http://127.0.0.1:8000/Dimmer1/set?Brightness:75

| | |
|---|---|
| '127.0.0.1' | The IP address of the computer running the master device representation being accessed, (likely to be different from localhost). |
| '8000' | The port number for the Web Server, as declared by the value of the „WEBSERVICES_PORT" constant created back in Step One. |
| 'Light1' 'Dimmer1' | The names of the smart devices being accessed, as declared when their representations were created in Step Two. |
| 'about' | A command to display general information about the specified device. |
| 'get' | A command to return the name of each state variable supported by the specified device, along with its currently recorded value. |
| 'set?Power:1' | A command to set state variable „Power" to a value of „1". Note: 0 = OFF, 1 = ON |
| 'get?Brightness' | A command to return the current value of state variable "Brightness". |
| 'set?Brightness:75' | A command to set state variable "Brightness" to a value of "75". |

### Creating a control point

An instance of the "ControlPoint" class allows external client programs to access devices via an EDK network. Figure 12 shows the API code required to perform this task. The String variable used in the "ControlPoint" constructor is a bespoke name for the specific instance being created. If more than one control point is being used within a client program, (i.e., to access different EDK networks), this variable can be used to identify specific instances.

A control point effectively acts as a portal into the EDK middleware system, allowing users to search for groups or specific master devices on the associated network. Its associated communications system provides details of the network which is accessed by a control point. In addition to providing details of an EDK network, the communication system supplied to a control point is also responsible for processing state update messages for the master devices. The control point itself automatically generates an internal "DeviceHub" instance, which is used to create and store clone device instances based on information received by the communications system from the EDK network. The control point accesses the stored clone devices and uses their information to provide returnable results for user searches.

### Searching for devices

To search for known master devices, present on an EDK network, a "ControlPoint" instance can use its "searchForDevices" method (Fig. 13). Instances of "ControlPoint" will only become aware of master devices upon receiving an update packet from them. Therefore, upon initially starting, the delay might occur before all master devices present on a network are discovered, concerning their update cycles when the control point joins the EDK multicast group. It is typically a good idea to enclose the search command in

```
InetAddress environmentAddress =
        InetAddress. getByName (NETWORK_ADDRESS);
Communications communications =
new Communications (environmentAddress, COMMUNICATIONS_PORT);

ControlPoint controlPoint =
        new ControlPoint (communications, "ControlPoint");
```

**Figure 12  Code for creating an EDK control point instance.**

```
Device[] deviceList = controlPoint.searchForDevices();
```

**Figure 13  Searching for known master devices on an EDK network.**

a "for", "do-while" or "while" loop to keep the control point scanning the network until a non-empty array is returned or the desired device is found. Alternatively, this command could also be repeatedly called from within an isolated thread, which runs continuously in the background, allowing devices that start broadcasting after the control point's initial search also to be detected.

### Searching for specific devices

If details of the target device or devices are known ahead of time, a client program can alternatively use one of the more specific search methods of the "ControlPoint" class. Figure 14 shows three such methods, each targeting different attributes of master devices. Firstly, the "SearchForDevicesByType" method can be used to filter the known list of master devices by their type, returning an array of any matching instances stored in the control point "DeviceHub". The topmost example in Fig. 14 uses this method to search for all instances of a "BooleanLight" The "Actuator" and "Sensor" classes in the API both contain numerous other declared variables that can be used with this method each representing one of the pre-formed smart device types included in the EDK. Alternatively, to search for a device type not included within the standard API, programs can use the "SINGLE_VARIABLE_DEVICE" and "MULTI_VARIABLE_DEVICE" variables, (also found in the "Actuator" and "Sensor" classes), or a bespoke device name entered as a String.

The two remaining search methods, shown in Fig. 14, are each designed only to return a single device, matching either a specified name or uuid criterion. If no matching device is found, then a "null" value is returned. If, for some reason, two different master devices existed on an EDK network and both were called "Light1", the "searchForDeviceByName" method will only return the first instance it encounters upon contents scanning of control point device hub. This also applies to the "searchForDeviceByUUID" method if both devices share the same aid value.[2]

### Processing EDK smart devices

The instance of "Device Hub" created by a control point is used to store clones created to represent networked master devices locally. The device hub creates the clones, which is the

[2] Note: When implementing master devices, it is highly recommended that each instance is given unique name value. It should always be ensured that device instances always have different uuid values, which can be generated randomly using the Java SDK UUID class.

```
// Search for all instances of a specific device type
Device [ ] deviceList = controlPoint.
        searchForDevicesByType (Actuator.BOOLEAN_LIGHT);

// Search for a single device with the specified name
Device device = controlPoint.searchForDeviceByName ("Light1");

// Search for a single device with the specified UUID
UUID uuid =
        UUID. fromString ("2deb19fc-5d41-4ac9-a0ca-4c6ff7a121e");
Device device = controlPoint.searchForDeviceByUUID (uuid);
```

**Figure 14 Three methods for finding specific devices.**

automatic response when information is received from the communication system that does not match any previously known representation. Clone devices are generally stored locally on the same computer as the control point used to create them. Aside from not possessing controllers for hardware, they are identical to the master devices that spawned them in every way. Even the unique attributes of the original master device are copied, including its name and quid value. As such, by using the associated classes within the EDK API, the details and states of each clone can be read, and in the case of actuators manipulated, in the same manner as if connecting directly to any master device instance.

The EDK API contains several different methods of reading the state of smart devices. A selection of the possible options is listed in Fig. 15. The determination of best method largely depends upon what information is required about the state of a device and how it is subsequently used.

From Fig. 15 examples, the topmost method is a general "getState" command, which is common to every EDK device. When called, the "getState" command will return a single string representation of the entire device, or more specifically, its state variable values. Responses are always sent in pairs, with the name of the state variable and its current value. For instance, in the case of the "Light1" "BooleanLight" device used in the implementation examples earlier, an "etState" request could result in either of the following String responses;

**Power : 0**
**Power : 1**

where, "Power" is the name of the only state variable included in a "BooleanLight" object, while zero (off) and one (on) are the current values of that state. The colon separating the two values acts as a key in a split command to allow easy separation of variable name and its value. In the case of multi-variable devices, which contain more than one state variable, an additional tilde key ("~") is added to separate the individual attributes. So, for the "Dimmer1" "DimmableLight" used in earlier examples, a "getState" request could return;

**Power : 0 ~ Brightness : 100**

where, "Power" is the first state variable and "Brightness" is the second, with current recorded values of zero (off) and one hundred (percentage of maximum illumination), respectively[3].

[3] Note: To prevent processing errors individual state variables should never include the colon or tilde characters (i.e.: or ~) in their names or possible returnable values.

```
String deviceState = device.getState();

String powerState = device.getState("Power");

int brightnessState =
        Integer.parseInt(device.getState("Brightness"));

int brightnessState =
        ((DimmableLight)device).getBrightness();
```

**Figure 15 Methods for reading the values of device state variables.**

```
for (Device device : deviceList) {
    if (device.getName().equalsIgnoreCase("Dimmer1") &&
            device.getType().equals("DimmableLight")) {
        if (device.getState("Power").equalsIgnoreCase("0")) {

            ((DimmableLight)device).turnOn();

        }

        else {

            ((DimmableLight)device).turnOff();

        }
    }
}
```

**Figure 16 An example of sending action state change requests to a master device.**

Figure 15 shows three methods that API can be used to return only the current value of a specific state variable, rather than the name/value pairs shown above. Generally, this is achieved by specifying the name of the state variable whose value is required as a variable in the method. The second and third down examples in Fig. 12 demonstrate this process for a "BooleanLight" and "DimmableLight" respectively. Typically, the returned state values are in a String format, but can easily be converted as shown with the "Brightness" variable example, where the value is converted into an integer once returned.

In many cases, the string can be avoided to integer conversion, as performed in the third example of Fig. 15. This is based on EDK API device's inclusion of convenience methods, which return state variable values in their most appropriate format automatically. This is demonstrated by the bottommost method in Fig. 15, which uses a "getBrightness" method found in the "DimmableLight" class, which automatically returns the state value as an integer. All that is required to use these bespoke convenience methods is to cast the generic "Device" object returned by a control point search into the appropriate device type class, as is shown in the example.

### Writing to smart devices

In the example provided by Fig. 16, an array returned by a control point search (as described in "Techniques/Protocols/Paradigms used in the Development of Proposed Work"), is scanned for a specific device called "Dimmer1", which is also a

```
package devicecontroller;

public class Controller {

    public Controller() {

    }

    public String getState() {

    }

    public String setState(String state) {

    }
}
```

Figure 17 A controller template for a single variable smart device.

"DimmableLight". If found, the value of the state variable "Power" is requested from the device. If the value returned for "Power" is zero (off) then a command is sent to turn the light on. For any other circumstance, the command is to turn the light off.[4]

In the example, the generic "device" variable taken from the "Device" array is cast into a "DimmableLight" object. This allows the programmer to access the two convenience methods "turnoff" and "turnOn," which are contained within the Actuator class. The EDK API contains several models for intelligent devices that can be used in place of the more generic actuator and sensor classes. Some of these classes also contain further convenience methods specific to that device type. For example, the "DimmableLight" class also contains a "setBrightness" method to allow a specific value to be entered for a "Brightness" state variable, controlling the amount of light emitted by the device.

### Implementing EDK compatible device controllers

Recalling back to "Step One: Declaring Constants", two of the constants that needed to be declared when implementing a master device (as shown in Fig. 5) were "CONTROLLER_CLASS" and "CONTROLLER_PACKAGE". The origins of the values used in the Fig. 5 example, can be seen in Fig. 17. The value that should be used for the "CONTROLLER_CLASS" constant is the name of the main class of the control system program, which in the example is simply "Controller". Additionally, the value for the "CONTROLLER_PACKAGE" is the name of the package containing the declared main class, in this case "devicecontroller".

When implementing a control program for smart devices to be used with an EDK middleware system, Fig. 17 shows the minimum classes required for integration. More specifically, the "getState" and "setState" methods are both essential and should be used to directly return or update the current state of the associated hardware, respectively. If additional code is required to create a link with the associated hardware, such as using a third-party software package (e.g., RXTX or a manufacturer API), then all this code should be placed into a constructor within the main class, as shown by the "Controller" constructor in Fig. 17.[5] If necessary, the constructor code should establish a connection

[4] Note: In the example provided in Fig. 16, "Actuator.DIMMABLE_LIGHT" could also have been used instead of the String value "DimmableLight."

[5] Note: Methods used in device controllers must return state values in as a single String using the expected EDK formats. It is recommended that the "setState" method returns the same value as an action request to "getState" once completing its operation.

```
package devicecontrollermultiple;

public class Controller {

    public Controller() {

    }

    public String getBrightness() {

    }

    public String getState() {

    }

    public String setBrightness(String value) {

    }

    public String setState(String state) {

    }
}
```

**Figure 18 A controller template for a dimmable light (multi-variable smart device).**

with the smart device hardware and then maintain it as a global variable that can be accessed directly by the "getState" and "setState" methods. Alternatively, the constructor could be used to start an isolated thread, which uses locally declared variables, also accessible by the "getState" and "setState" methods, to handle state action requests. It is essential that no code necessary to directly establish a connection with hardware is included in either the "getState" or "setState" method as doing so could lead to overall instability in the EDK middleware system.

### Controllers for multi-variable devices

Figure 18 highlights how the controller code in Fig. 17 can be extended to handle smart devices with multiple state variables. The code presented (in Fig. 17), could function with a multi-variable smart device, but it is often desirable to separate certain state variables to better structure program code, or for efficiency, etc. Figure 18 shows a template for the controller used with the "DimmableLight" device "Dimmer1" example mentioned throughout this guide. Notice how the "getState" and "setState" methods have been retained (for handling the "Power" state variable), although the "Brightness" state variable has been separated and given its handling methods, namely "getBrightness" and "setBrightness." To add additional "get" and "set" methods, it is necessary to ensure that their suffix is named the same as the state variable they are expected to handle, (e.g. "getPower", "getBrightness", "setPower", "setBrightness", etc.).[6]

When accessing a loaded device controller, an EDK implementation will first search through the methods of the declared controller class (i.e. "CONTROLLER_CLASS") to see if it contains a bespoke match for the current state variable it needs to process. If no appropriate method matching the state variable can be found, the system will then

[6] Note: Any "set" methods should also only expect to receive a single variable containing the new state value to be processed, as shown in Figs. 17 and 18. No other variables must be added for the methods to be compatible. Note: As with "setState" it is recommended that any "set" method returns the same value as an action request to "getState" once completing its operation.

automatically default to either the "getState" or "setState" method, depending upon which action is being performed.

Additionally, in Fig. 15, the package name has changed, reflecting that it is a different device controller program from Fig. 14 example. To be loaded correctly into the EDK and used with a master device, Fig. 15 example would need to specify "devicecontrollermultiple" as the value for "CONTROLLER_PACKAGE" during the first implementation step.

# EVALUATION

## Evaluation strategy

One of the criteria used to measure the success of the EDK is its effective processing of action requests for smart devices, compared to existing middleware solutions. Since UPnP and Web Services largely inspired the new mechanism, examples of these architectures are used as a benchmark. Additionally, EDK testing with sensors and actuators is necessary, to acquire accurate results for both sending and receiving state variable data. Therefore, the middleware was integrated into the control systems of two different intelligent environments, located at the University of Essex, namely the iClassroom and the iSpace test-beds.

The intelligent environments chosen for the evaluation were selected as benchmarks to take advantage of the established middleware in their existing control systems. During the evaluation, EDK and benchmark middleware systems were tested by controlling a DMX-based dimmable spotlight and a Phidgets light sensor. Primarily, it assesses the average processing speed required by the different components of the EDK (i.e., API control and Web Services) compared to that needed by the equivalent benchmark systems. If the EDK was capable of performing the same tasks as an existing middleware system in approximately the same time frame, it is considered a viable alternative for use in an intelligent environment control system. Incidentally, it was anticipated that the EDK would actually require a significantly lower processing time than UPnP or Web Services, mostly due to the greater emphasis on locally processed variables.

## Evaluation results

To evaluate the EDK middleware system, each mechanism was tested separately, with no other programs running at the time. For each middleware system, five hundred 'get' or 'set' requests were made to the same sensor or actuator, respectively. This workload was split across five sessions of one hundred requests each. To keep the test fair, each session was performed using the same computer, which ran both timer program measuring the processing intervals, plus made the state requests to the relevant middleware implementation.

### System validation: sensors

The first experiment in the project evaluation focused specifically on comparing middleware used to access sensors. In other words, devices whose functionality consisted only of sending information about their current state back to a requesting program. The

**Table 1  Average middleware processing times for taking a reading from a light sensor.**

| | Average Processing Time Required (milliseconds) | | | | |
|---|---|---|---|---|---|
| | Session 1 | Session 2 | Session 3 | Session 4 | Session 5 |
| Jetty-based Web Services | 80.03 | 77.43 | 55.58 | 61.48 | 60.65 |
| Youpi UPnP | 52.51 | 35.51 | 99 | 22.01 | 106.56 |
| Web Services (EDK) | 1.99 | 1.66 | 1.61 | 1.64 | 1.40 |
| Direct API Control (EDK) | 0.00542 | 0.00532 | 0.00516 | 0.00517 | 0.00519 |

Jetty-based Web Services system in the iClassroom linked directly into individual device control programs, which automatically requested a current reading from the relevant sensor hardware when called. However, the iSpace UPnP mechanism functioned in a very similar style to the EDK middleware design. Specifically, whenever called, the 'getState' command would access the control system for a specific device and return the last recorded value obtained from the actual sensor itself. A 'getState' method call never actually resulted in the hardware being accessed directly in order to take a reading. Updates to the recorded light level value were handled by an independent thread located within the program code for the sensor, which automatically took a new reading from the hardware intermittently. In the case of the iSpace UPnP system, a new reading was recorded approximately once every ten seconds. For EDK, this interval was reduced to approximately three seconds, in order to reduce the margin of error between the recorded and actual sensor values.

The experiment involved measuring the time required for each middleware system to return a lux value for a specific light sensor Phidget. As can be seen from the results displayed in Table 1, on average for each session run, both device interface methods offered by the EDK middleware required significantly less time to process individual action requests than both the Jetty-based Web Services or Youpi UPnP implementations. In the case of the Direct API interface method, the processing time was reduced to nanoseconds as the sensor value being returned in response to each action request was taken from a locally stored variable within the clone of the real sensor generated by the EDK. As mentioned above, the Youpi UPnP middleware used a similar method, but the recorded values for the light sensor were stored on a remote server, hence required additional processing time for the system to access and get the data values from. Unlike the API interface method, action requests made using the EDK Web Services were directed at the real light sensor, yet they still required a fraction of the processing time of the Jetty-based system, consistently throughout the evaluation.

### System validation: actuators

The second experiment performed for the evaluation focused on actuators. More specifically, this involved devices that contained one or more variables that could have their state modified, via the middleware accessing an attached control system. When setting the state of a variable, the success of the corresponding hardware being updated was not always guaranteed. This is because many devices do not provide any feedback to

**Table 2 Average middleware processing times changing the brightness of a dimmable light.**

| | Average Processing Time Required (milliseconds) | | | | |
|---|---|---|---|---|---|
| | Session 1 | Session 2 | Session 3 | Session 4 | Session 5 |
| Jetty-based Web Services | 127.09 | 133.46 | 112.57 | 113.92 | 118.29 |
| Youpi UPnP | 171.36 | 232.33 | 212.22 | 238.08 | 180.49 |
| Web Services (EDK) | 101.83 | 102.45 | 102.35 | 102.26 | 102.09 |
| Direct API (EDK) | 0.67 | 0.64 | 0.60 | 0.59 | 0.55 |

the controller code, concerning whether a transmitted command has been received or acted. Both, Jetty-based Web Services and Youpi UPnP middleware sent acknowledgments back to the control programs where action requests originated but always assumed a positive outcome. In contrast, the EDK did not send any acknowledgments in direct response to individual state change requests. The success or failure of an update could be determined by observing the next set of variable settings transmitted by a real device to its clones.

This experiment involved each middleware system accessing a single DMX-controlled dimmable spotlight and alternating its brightness level between fifty and one hundred percent. Table 2 lists the average processing times required by each batch of one hundred action requests made during the experiment. As with the sensor experimentation, both EDK interface mechanisms were found on average to require significantly less processing times than the Jetty-based Web Services and Youpi UPnP systems. It is worth noting that the difference between the EDK and Jetty-based Web Services was not as profound as for the sensors. This was due to the controller code for the DMX lights adding a delay of approximately one hundred milliseconds to the processing of each action request via a compulsory sleep command. As the EDK Web Services were directed at the representation of the actual light rather than a local clone, the delay was unavoidable.

Unlike for the sensor 'getState' action requests, when setting a device's state, the EDK's Direct API interface method also targeted the original 'master' representation rather than a clone. Effectively, a state change action request was forwarded to the original device by sending out a state change request using the inbuilt multicast communication system. As the system did not require any acknowledgment, the process could be ended at that point, unlike UPnP and the Web Services mechanisms, which required some kind of response, even if based upon an assumption.

## Evaluation summary

Concerning the processing times for action requests, on average, in every session of the evaluation, both interface mechanisms of the EDK were found to be significantly superior to the benchmark Jetty-based Web Services and Youpi UPnP middleware systems. This result was not entirely unexpected as the EDK utilised locally stored variables much more than either of the benchmark systems. Furthermore, in cases where remote access is necessary, (i.e. when setting the state of an actuator), the EDK does not require any

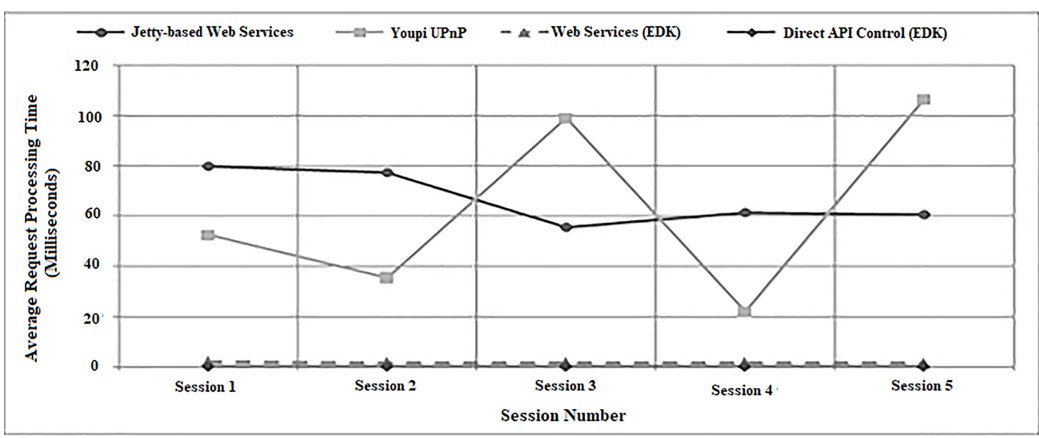

**Figure 19 A graph showing average processing times for setting a light (experiment one).**

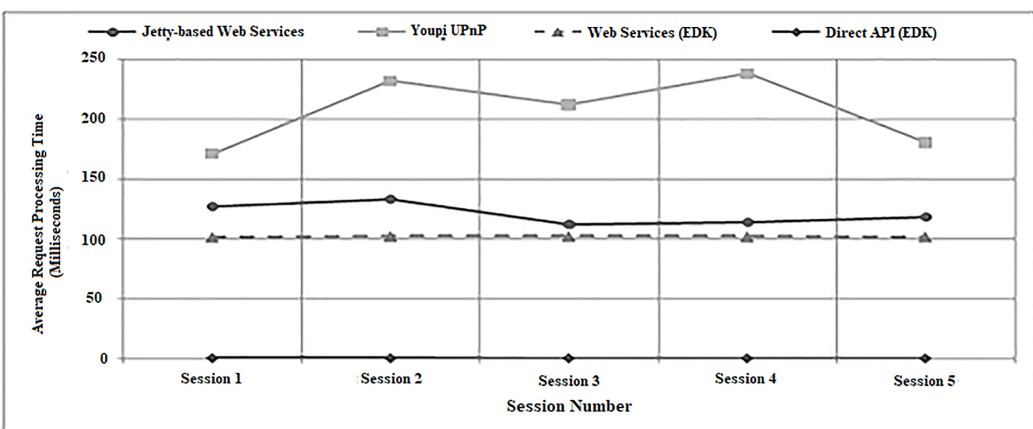

**Figure 20 A graph showing average processing times for setting a light (experiment two).**

confirmation of a device receiving an action state request to be returned, further reducing the processing times.

For experiment two, the additional 'power' state variable of the DMX spotlight was manually set to an 'on' state prior to each experiment session to observe the brightness changes but was otherwise unused. However, it should be noted that for each action request sent to the UPnP middleware system, it was necessary for its control point to first perform a search of all the DMX light state variables in order to isolate the one controlling the brightness before the new state could be set. As more than one state variable existed within the device (i.e., power and brightness), this may have caused the action request processing time to be extended, depending upon which was discovered first. This additional step was not necessary for the EDK or Jetty-based Web Services, where the individual state variables were already declared within the device controller code. This may explain why the UPnP mechanism required more time than any other mechanism in

each session when handling an actuator, despite being faster than Jetty-based Web Services in three out of five sessions during the earlier sensor experiment.

Finally, another observation made during the evaluation was that for each batch of one hundred action requests performed, while the average processing times generated by Jetty-based Web Services and Youpi UPnP fluctuated noticeably between sessions, the times for both EDK methods remained largely static. This effect is illustrated by the graphical representations of the results from the evaluation presented in Figs. 19 and 20. These figures also clearly highlight how the EDK was capable of outperforming the two benchmark middleware mechanisms consistently throughout the entire evaluation.

## CONCLUSIONS

This paper discussed recent work performed into creating a new middleware architecture for an intelligent environment control system. It shared insights on how control system developers often require some functionality trade-off when integrating one of the numerous existing architectures. Alternatively, an existing mechanism may be manually augmented with additional code to overcome any missing functionality, which could result in detrimental synchronization issues within the resulting control system. This project aimed to create a new middleware, capable combining the positive features of two well-known architectures (namely UPnP and Web Services) into a single mechanism, to overcome functionality and synchronisation issues. Most importantly, it envisaged that the new mechanism would be required to process state action requests as quickly as the two benchmark middleware architectures.

Complex, intelligent environment developers must implement their additional bespoke layer of middleware to link devices and services provided by different manufacturers. iClassroom and iSpace control system architectures in two full-scale intelligent environments are also presented.

It also discusses the Environment Development Kit (EDK) as the new alternative middleware as an outcome of the research performed by this project. It assesses how viable the new middleware design was as an alternative to UPnP and Web Services. This strategy involved two separate experiments testing sensors and actuators independently due to their different properties. It showed that a comparative analysis of the average action request processing times required by the EDK, plus implementations of the benchmark UPnP and Web Services systems.

The evaluation experiments both indicated overwhelmingly that the two interface methods offered by the EDK architecture implementation could process state action requests to sensors and actuators faster than either benchmark mechanism. Out of the ten sessions performed, either benchmark did not outperform the EDK in a single case. EDK and UPnP used stored variables when returning sensor state data, rather than accessing the actual device directly. These stored variables were updated periodically by an independent thread, which noted the actual sensor readings. However, in the UPnP implementation, a ten-second delay between individual readings was observed along with an update of the stored variable. For the EDK implementation, this delay was reduced to three seconds, yet the mechanism was still capable of processing action requests faster

than the UPnP benchmark. The results strongly indicated that the EDK is a viable alternative to UPnP and Web Services in a smart device or intelligent environment control system.

Another important issue for EDK was to resolve the data de-synchronised using multiple device interfacing methods. As the new mechanism was designed from the ground up, steps could be taken to allow synchronisation between the API and Web Services interface methods to be practically guaranteed. For this, both interface methods use the same sets of variables to store descriptions of individual master or clone devices. Also, clone devices were regularly synchronised with their original master; locally stored representations were constantly being updated with the known true state values. This helped defense against de-synchronization of data stored in clone devices.

### Future work

With the initial framework for the EDK established, future researches can expand the existing middleware architecture to integrate additional functionality. Firstly, an internal mechanism can be introduced to add an internal authentication system, which allows instances of smart devices represented by the mechanism to optionally be assigned with a password. Unless the correct password was supplied during communication, neither control points nor other interfaces could access the device, nor a clone of the original be created as a result of a search request or update processes.

Therefore, future study should focus on EDK development, where its variables and methods allow control points and other interfaces to identify the specific environment where an instance of a smart device is located. Technically this could already be achieved to some extent using the existing mechanism, for example, by manipulating the name variable when creating device instances. However, with the augmentations would allow the handling of scenarios such as the device changing locations to a different intelligent environment. It would also make it easier to represent multiple different environments on the same network. This could potentially allow the creation of agents or control programs that use devices from several different environments collaboratively, as part of their functionality.

## ACKNOWLEDGEMENTS

The author is highly grateful for all the associated personnel who contributed in the completion of this Scale Up Project.

### Funding

The authors received technical and financial support from the Deanship of Scientific Research (DSR) at King Abdulaziz University, Jeddah, under Grant No. (RG-7-611-39). The funders had no role in study design, data collection and analysis, decision to publish, or preparation of the manuscript.

## Grant Disclosures

The following grant information was disclosed by the authors:
King Abdulaziz University, Jeddah: RG-7-611-39.

## Competing Interests

Daniyal Alghazzawi, James Dooley, Dr. Ghadah Aldabagh & Dr. Abdullah Alghamdi declare no conflict of interest.

## Author Contributions

- Daniyal Alghazzawi conceived and designed the experiments, analyzed the data, performed the computation work, prepared figures and/or tables, authored or reviewed drafts of the paper, and approved the final draft.
- Ghadah Aldabbagh analyzed the data, performed the computation work, prepared figures and/or tables, authored or reviewed drafts of the paper, and approved the final draft.
- Abdullah Saad AL-Malaise AL-Ghamdi analyzed the data, performed the computation work, prepared figures and/or tables, authored or reviewed drafts of the paper, and approved the final draft.

## Data Availability

The code is available at figshare: Alghazzawi, Daniyal; Dooley, James (2020): Environment Development Kit Middleware. figshare. Software. https://doi.org/10.6084/m9.figshare.12894440.v1.

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
