# Peer review of "ScaleUp: middleware for intelligent environments"

_PeerJ Computer Science, doi:10.7717/peerj-cs.545_

## Round 0.1 · original submission · Major Revisions

Paper is interesting but the reviewers opinion is that it requires further elaboration before being considered for publication. Essentially it misses a more scientific description of the system presented. I suggest to change figures for diagrams.

Reviewer 1 ·

Basic reporting

The authors present a paper called “ScaleUp: Middleware for intelligent environments” that focuses on its usage for the Internet of Things. It is the opinion of this reviewer that it requires further elaboration before being considered for publication:

1. The authors mention “the developed mechanism serves as a viable alternative to the commonly used Universal Plug and Play (UPnP) and Web Services middleware systems”, which is quite a statement since WS and UPnP have been standards in industry and research for decades. At this stage of their middleware development, it is hard to make such a claim, so the authors should humble down this sentence.
2. Line 74 “For instance, if a device state variable, updated using one middleware system, was not reflected by all the alternative mechanisms used”. This looks like an incomplete sentence that should be finished.
3. The references provided in literature could be expanded with usage of middleware in other international research projects. A (non-exhaustive) list of references that could be added is : A Review of Middleware for Networked Robots, Middleware Architectures for the Smart Grid: A Survey on the State-of-the-Art, Middleware Technologies for Smart Wireless Sensor Networks towards Internet of Things: A Comparative Review, Taxonomy and Main Open Issues, An optimized, data distribution service-based solution for reliable data exchange among autonomous underwater vehicles, Maritime Data Transfer Protocol (MDTP): A Proposal for a Data Transmission Protocol in Resource-Constrained Underwater Environments Involving Cyber-Physical Systems or IoT Middleware: A Survey on Issues and Enabling Technologies. Of course, these are suggested just as a means to improve the paper, they are not mandatory to be included if the authors think otherwise and provide other similar ones.
4. The middleware architecture that has been created lacks a specific name, it is only referenced as part of the project (ScaleUp). That makes hard to understand the differentiating value of this architecture compared to the other.
5. It is not clear what the advantages of the middleware architecture are compared to the existing literature. An open issues section, where it is mentioned what open issues are being tackled by the proposal, must be added. Likewise, the contributions made by the proposal should be made clear in the manuscript in the first section.
6. There is very little information about fundamentals of middleware: a) is a Message-Oriented Middleware? A Middleware architecture? Or something else? b) How do the PDUs interchanged look like (if it applies)? c) What software components are included in the MW? d) What software layers are used within the middleware itself (Hardware Abstraction Layer, Service Application Points, etc.)? All these questions should be answered thoroughly.

Experimental design

It is detailed enough.

Validity of the findings

They are suitable for the topic of the journal

Additional comments

No comment

Reviewer 2 ·

Basic reporting

Paper is well presented, but I think it needs really hard work to Abstract
Inline 41 authors present a “new form”, in line 45, authors present a “new mechanism” and in line 49, authors present a “system”. It is necessary to concrete the product, or solution, presented in the paper. Title names a middleware, it is necessary to concrete the contribution of the paper.
Line 43: UPnP is commonly considered a protocol instead of a middleware, perhaps it is necessary
Lines 47 (end) and 48: "...types of the device...” change for “...types of devices...”
Line 49: authors relate a “clone devices” that implies that the system presented uses clonation, this aspect must be presented previously.
Line 50: the same of the previous comment, but about the concept of “master”.

Abstract uses specifically concepts that, probably, paper can clarify, I suggest rewriting the abstract.

Introduction
Line 57: I suggest to change “omnipresent” for “necessary” or “ubiquitous”, instead omnipresent is correct, it is more frequently to use the alternatives mentioned above.

The introduction clarifies the aims of the research. I miss a paragraph with the description of the paper structure (forget this comment if this paragraph is not mandatory by the journal guide)

Literature review
Line 109: instead of “bespoke” is correct, I suggest to use the more frequently expression “custom-made” or “proprietary” when the concept is used in commercial trades (Apple or Sony mentions).
Lines 124 and 125 present the interesting result that can be used in the paper (requirements), I suggest authors select from the 12 requirements which ones accomplish the system presented (for the sake of providing scientific validity to the work presented).

Line 183 (figure 1): If the objective of the figure is to demonstrate where the wrapper should be inserted, a figure is not necessary, it is enough to explain in the text the sequence between the device and the programming interface. However, I suggest extending the figure with the context of layers in which a middleware acts.

Line 184: the reference Doolet et al., 2011 probably is the reference Dooley et al, 2011

Line 189: I try to avoid referring to tools by means of URLs, I suggest looking in PeerJ or similar for any reference to a survey in which the use of Eclipse as a system is mentioned.

Line 190 (figure 2): I don’t know the significance of the home environment pictures related to the blocks diagram used in the figure.

Figures 1, 2, and 3 represents the role of the main blocks, or functionalities, offered between devices and interface programs. It could be interesting to unite them in a comparative figure where device and programming interfaces were common and the different ways of implementing middleware could be compared.

The Environment Development Kit (EDK)
Figure 4: a code snippet can be useful in an implementation section, but a code declaration does not seem convenient in this section. In the text, authors declare that figure 4 represents an architecture, but no code.

Line 258: an example of a middleware based on publish-subscribe can be found in this reference (I suggest to include in the paper):
“Simó-Ten, J. E., Munera, E., Poza-Lujan, J. L., Posadas-Yagüe, J. L., & Blanes, F. (2017, June). CKMultipeer: connecting devices without caring about the network. In International Symposium on Distributed Computing and Artificial Intelligence (pp. 189-196). Springer, Cham.”

In general, to understand the next lines about the different operations offered by the EDK, I really miss some figures (instead of the code fragments presented). These figures are:
A general figure, using block diagrams or similar methods, to represent the different elements of the middleware (devices, master, clones, etc.). This figure can be used to explain the different functionalities presented.
A figure (as a flow chart or UML sequence diagram) to each process: Implementing master devices, Searching for devices, Searching for specific devices, etc.

Line 431: searching for devices, perhaps is a discovering mechanism.

Additionally, authors can present the operations in a section that describe the middleware and present a case of use as an example that describes the diagrams before mentioned.

Evaluation
I suggest renaming the section “Experiments” or “System validation”.
Experiments are well presented, I only miss the best quality in figures 19 and 20.

Conclusions and future work
Well presented and really concise and concrete.


Authors mentioned in lines 84 and 85 that the work is based on the project (Scale up) but this project is not mentioned in acknowledgements or Founding.

Experimental design

The experimental design is well done, is based on the comparison with other systems. Is not mandatory, but I suggest to compare other scenarios including different devices.

Validity of the findings

no comment

Additional comments

no comments

---

## Round 0.2 · accepted · Accept

As you can read below, Reviewer #1 is happy with the changes and effort that you have put into the revised version.

Reviewer 1 ·

Basic reporting

The document has solved the issues that were mentioned before. I believe it is ready for publication. Usage of English is correct, background is provded correctly, the structure and notation are fine, it is relevant to the hypothesis that is put forward and terms are well-defined.

Experimental design

It was already good enough in the first review of the paper, so I believe it is still good.

Validity of the findings

Although limited, impact is real and significant. Scientific community can certainly benefit from innovative middleware ideas in the IoT. Required data have been provided. Conclusions are well defined.

Additional comments

The document has solved the issues that were mentioned before. I believe it is ready for publication.